# ATAC-Seq and RNA-Seq Integration Reveals Chromatin Accessibility and Transcriptional Dynamics During Fruit Color Development in Mulberry

**DOI:** 10.3390/ijms27010456

**Published:** 2026-01-01

**Authors:** Yichun Zeng, Yilei Wu, Jie Dai, Jiang Liu, Ling Wei, Sanmei Liu, Gang Liu, Gaiqun Huang

**Affiliations:** 1Sericulture Research Institute, Sichuan Academy of Agricultural Sciences (Institute of Special Economic Animal and Plant, Sichuan Academy of Agricultural Sciences), Nanchong 637000, China; 2Sichuan Academy of Agricultural Sciences, Chendgu 610066, China

**Keywords:** mulberry, transcription factors, ATAC-seq, RNA-seq

## Abstract

Fruit color, a defining characteristic in mulberry, varies markedly across cultivars, ranging from black and red to pink and white. However, the regulatory mechanisms governing pigmentation during fruit development remain poorly characterized. In this study, ATAC-seq and RNA-seq were integrated to investigate genome-wide chromatin accessibility and identify putative transcription factors involved in fruit development in Da10 (Yueshenda10, *Morus atropurpurea* Roxb.). Differentially accessible chromatin regions and differentially expressed genes were jointly analyzed to generate a genome-wide map of regulatory elements and transcriptional activity. This integrative approach enabled the reconstruction of a transcriptional regulatory network that facilitated the identification of candidate transcription factors and target genes associated with the biosynthesis of bioactive compounds, including sugars, flavonoids, diterpenoids, and anthocyanins. The expression levels of RNA-seq were confirmed by qRT-PCR. Motif enrichment analysis revealed seven and 23 enriched cis-elements at the S1 and S3 developmental stages, respectively. Integration with the transcriptional regulatory network established a robust framework for identifying candidate genes involved in the accumulation of key metabolites. Additionally, the dataset enables exploration of previously uncharacterized traits during turning stages of mulberry fruit development. Collectively, these findings advance our understanding of the regulatory architecture underlying fruit maturation and provide a foundation for future strategies to enhance mulberry fruit quality.

## 1. Introduction

Mulberry (*Morus alba* L.) is among the earliest plant species officially recognized in China for its dual medicinal and edible qualities [1,2]. Accumulating evidence has highlighted its broad spectrum of health-promoting benefits, including cholesterol-lowering, anti-obesity, hepatoprotective, vision-enhancing, and joint-supporting effects, primarily attributed to its high content of bioactive compounds, including anthocyanins, alkaloids, and flavonoids [2,3]. With rising consumer awareness of these functional properties, there is increasing demand for improved fruit quality encompassing texture, flavor, and nutritional value [1,3]. Fleshy fruit development involves tightly coordinated physiological transitions, such as changes in color, firmness, nutrient contents, and metabolic composition, driven by complex networks regulating the biosynthesis of sugars, organic acids, and pigments such as anthocyanins [1,4,5], or carotenoids in case of tomato [6].

Despite growing interest, research on the molecular basis of these processes in mulberry has remained limited. Most previous studies have focused on a narrow set of structural genes or transcription factor involved in metabolic biosynthesis, offering an incomplete view of regulatory dynamics [7,8]. Transcriptomic profiling has identified associations between fruit ripening and specific transcription factors (TFs), such as ethylene-responsive factors in the AP2/ERF family, which influence pathways related to sucrose metabolism, anthocyanin accumulation, and cell wall modification [4]. However, the role of chromatin accessibility in regulating these transcriptional changes during mulberry fruit maturation remains poorly understood.

Chromatin accessibility reflects the extent to which chromatin structure permits regulatory elements to interact with DNA, thereby influencing gene transcription. Binding of TFs typically promotes transcriptional activation by disrupting nucleosome assembly at key regulatory regions such as promoters, enhancers, insulators, and locus control regions [9]. Mapping the open chromatin landscape is therefore essential for elucidating the transcriptional regulatory framework underlying fruit development. Key TFs implicated in sugar signaling—WRKY-domain proteins, MYB factors, ABA-responsive element binding proteins (AREB) and basic leucine-zipper (bZIP) proteins—have been characterized in apple [10], tomato [11] and maize [12]. Likewise, the MYB–bHLH–WD40 triad that governs anthocyanin accumulation in blueberry has been described [13]. Nevertheless, the molecular regulatory mechanisms by which TFs control the biosynthesis of sugars and anthocyanins in mulberry remain largely unresolved. Assay for Transposase Accessible Chromatin using High-Throughput Sequencing (ATAC-seq) offers a powerful tool for studying chromatin accessibility at high resolution across specific developmental stages [14]. The application of ATAC-seq in diverse plant systems, such as areca palm [15], tomato [6], and rice [16], maire yew [17], has demonstrated its utility in mapping transcriptional regulatory landscapes. By overlaying RNA-seq transcriptomes on ATAC-seq chromatin maps, we can systematically identify stage-specific enhancers and their cognate transcription-factor–target-gene networks.

In this study, ATAC-seq and RNA-seq were integrated to investigate chromatin accessibility dynamics and reconstruct transcriptional regulatory networks during two key transitional stages (S1 and S3) of mulberry fruit development. This integrative analysis generated a comprehensive dataset that enabled the identification of candidate TFs and their putative targets involved in the regulation of key quality characteristics. These findings provide a molecular framework for understanding fruit maturation and offer a basis for the targeted improvement of mulberry fruit quality.

## 2. Results

### 2.1. Determination of Anthocyanin Content in Different Fresh Fruits

To elucidate the molecular mechanisms underlying anthocyanin accumulation, anthocyanin content was quantified in fresh fruits from 60 mulberry varieties (Appendix A). Among them, Da10 (Yueshenda10) was distinguished by its high content of bioactive compounds and favorable commercial traits. Consequently, Da10 was selected as the model cultivar for subsequent analyses.

### 2.2. Landscape of Accessible Chromatin Regions

A total of 63.11 Gb of sequencing data were generated from four fruit samples collected at 10 (S1) and 19 (S3) days after flowering, with each contributing at least 15.07 Gb. Following preprocessing and quality control, 43.61 Gb of clean reads were retained, with individual samples yielding between 10.47 Gb and 11.76 Gb. The Q30 base percentage exceeded 92.87% across all samples (92.65–93.25%), indicating high data quality suitable for downstream analyses (Appendix A). Open chromatin signals were obtained through read alignment, deduplication, and peak calling. Clean reads were mapped to the *M. atropurpurea* Roxb. reference genome, with mapping rates ranging from 86.03% to 90.76% (Appendix A). After removal of duplicate reads, coverage was uneven across chromosomes but consistent between positive and negative strands (Appendix A). Peak calling identified 5776 (S1_1), 5993 (S1_2), 12,609 (S3_1), and 16,129 (S3_2) peaks, and the fraction of reads in peaks (FRiP scores) ranged from 0.38 to 0.80 (Appendix A). TSS enrichment was evident in all samples, reflecting high ATAC-seq data quality (Appendix A). Chromatin accessibility was notably higher in S3 compared to S1.

To assess peak reliability, the IDR approach was applied using pseudo-replicate peak sets. Approximately half of the total peaks were classified as reproducible. Peaks passing the IDR threshold (≤0.05) were retained for further analysis. The mean width of high-confidence peaks was 0.40 kb in S1 and 0.44 kb in S3 (Appendix A). TSS enrichment was observed in both groups, with S1 peaks exhibiting a sharp, narrow TSS signal with low variability and S3 peaks displaying a broader and less defined TSS enrichment with wider confidence intervals (Figure 1a). To identify genes associated with transposase hypersensitive sites (THSs), the nearest gene to each peak was annotated. A total of 2590 and 3331 THS-associated genes were identified in the S1 and S3 groups, respectively, with many genes associated with multiple peaks. ATAC-seq signal intensity gradually decreased with increasing distance from the TSS, although a subset of accessible regions persisted beyond 10 kb from the TSS (Appendix A). Promoter regions accounted for 53.82% and 33.37% of peaks in S1 and S3, respectively (Figure 1b). Approximately 50% of the accessible regions were located outside promoter regions, suggesting the presence of other functional elements. Notably, distal intergenic regions were significantly more prevalent in the S3 samples (Figure 1c).

To further evaluate differential chromatin accessibility, ATAC-seq profiles were compared between S1 and S3 stages. Principal component analysis (PCA) demonstrated a strong correlation between biological replicates within each group (Appendix A). In total, 173 DARs were identified based on the thresholds of log2 (fold change) ≥ 1 and FDR ≤ 0.05. Notably, all DARs were up-regulated (Figure 1d).

### 2.3. Functional Analysis of the Genes Related to Chromosome Open Region

To investigate the regulatory functions of accessible chromatin, functional enrichment analysis was performed on genes adjacent to THSs and DARs. In total, 363 (9.8%) S1-specific genes, 1104 (29.9%) S3-specific genes, and 2227 (60.3%) shared genes were identified (Figure 2a). THS-associated genes in S1 and S3 were mapped to distinct genomic features in the *M. atropurpurea* Roxb genome. Among S1-specific genes, eight (11.9%) were located in 3′ UTRs, 112 (6.6%) in distal intergenic regions, two (7.1%) in downstream regions, nine (9.5%) in exons, 22 (6.8%) in introns, and 246 (13.9%) in promoters. Corresponding S3-specific genes included 13 (19.4%) in 3′ UTRs, 712 (41.8%) in distal intergenic regions, four (14.3%) in downstream regions, 26 (27.4%) in exons, 129 (39.8%) in introns, and 393 (22.2%) in promoters (Appendix A). Among them, 173 S3-up-regulated DARs mapped adjacent to 92 genes.

KEGG pathway enrichment analysis revealed the S1-specific genes were related to plant hormone signal transduction, biosynthesis of various nucleotide sugars, phosphatidylinositol signaling system, glycosyltransferases, ribosome, and pentose and glucuronate interconversions, suggesting roles in early plant growth and developmental processes (Appendix A). Combined analysis of up-regulated DARs and THSs was performed to define the S3-specific gene set. KEGG pathway enrichment revealed that the resulting 1 196 genes were associated with basal transcription factors, Ras signaling, diterpenoid biosynthesis, glucosinolate biosynthesis, valine, leucine, and isoleucine biosynthesis, DNA repair and recombination, sesquiterpenoid and triterpenoid biosynthesis, isoflavonoid biosynthesis, prenyltransferase activity, transcription machinery, and stilbenoid, diarylheptanoid, and gingerol biosynthesis (Figure 2b).

### 2.4. Motif and Footprinting Analysis of DARs in ATAC Peaks

Motif analysis of S1 DARs was performed on 375 sequences, resulting in the identification of 441 motifs, which were classified into seven unique motif types (Appendix A). Sequence alignment against motif databases, including non-redundant plant motifs, JASPAR, TAIR, and PlantTFDB, revealed the presence of 246 TCP, 93 HMG, 76 bZIP, 59 FAR1, and 44 B3 TF-binding motifs. Based on *M. atropurpurea* Roxb genome annotation, the chromosomal distribution of these motifs was visualized (Figure 3a). Motifs were unevenly distributed across the 14 chromosomes, and no correlation was observed between motif number and chromosome size. For example, the largest chromosome (LG1) contained only 17 motifs, while the smallest chromosome (Chr 9) contained 40 motifs.

A total of 1839 DARs from S3, including 173 up-regulated and 1666 stage-specific peaks, were also subjected to motif analysis, yielding 4443 motifs classified into 23 unique motif types (Appendix A). These motifs varied from 6 to 50 nucleotide acids in length. Alignment with known motif databases identified 1014 NAC, 793 MYB, 713 AP2/ERF, and 413 TCP motifs. Chromosomal distribution analysis revealed substantial variation across the 14 chromosomes (Figure 3b). The number of motifs on each chromosome was not correlated with chromosome size. For example, the lengths of LG3, LG4, LG5, and LG6 are the same, while LG4 contains more motifs than the others. Furthermore, LG6 was the longest chromosome among LG6 to LG14, while the number of motifs in LG6 was the least.

While MEME identifies conserved sequence motifs by comparing DAR sequences to background genomic DNA, it does not localize transcription factor binding sites or quantify motif activity. To address this limitation, footprinting analysis was conducted on motifs within both up-regulated and stage-specific DARs to detect putative TF binding sites and infer direct occupancy. Footprint scores were consistent between biological replicates of S1 and S3 (Appendix A). However, a significant difference in footprint scores was observed between the two developmental stages (Appendix A). Motifs with an absolute log2 (fold change) in footprint scores ≥ 1 were considered to exhibit condition-specific binding activity.

### 2.5. Integrated Analysis of ATAC-Seq and RNA-Seq Data Revealed Chromatin Accessibility Regulating Gene Expression

The pronounced differences in chromatin accessibility between S1 and S3 prompted further investigation into the transcriptional regulatory landscape. To explore DAR-mediated transcriptional regulation during Da10 fruit development, RNA-seq data for S1 and S3 stages in three biological replicates were downloaded from the NCBI database and processed bioinformatically (Appendix A). A total of 40.46 Gb of clean data were obtained across six samples, with individual samples yielding between 5.90 Gb and 8.48 Gb and Q30 base percentages exceeding 93.47% (92.25–93.59%), indicating high sequencing quality. Clean reads were mapped to the *M. atropurpurea* Roxb reference genome with mapping efficiencies ranging from 96.29% to 96.85% (Appendix A). Gene expression levels were quantified based on raw read counts per gene. PCA revealed that the first two principal components (PCs) explained 97.68% of the total variance, with clear separation between S1 and S3 samples (Figure 4a). Differential expression analysis identified 6 186 genes showing significant changes between the two stages, including 1924 up-regulated and 4262 down-regulated genes (Figure 4b).

GO enrichment analysis of down-regulated genes in S3 indicated significant enrichment in photosynthesis and secondary metabolic processes, including photosynthesis, light harvesting in photosystem I, photosynthesis-light harvesting, photosynthesis-light reaction, and secondary metabolic process in the BP category, and monooxygenase activity, oxidoreductase activity, and pigment binding and chlorophyll binding in the MF category (Appendix A). KEGG pathway enrichment analysis showed similar patterns (Appendix A). For the up-regulated genes in S3, GO terms were enriched in nucleoside metabolism and ATP-related processes (Appendix A). Additionally, KEGG analysis revealed that pathways associated with anthocyanin and carbohydrate metabolism such as glycolysis/gluconeogenesis, flavonoid biosynthesis, glycine, serine, and threonine metabolism, anthocyanin biosynthesis, and tyrosine, phenylalanine, and tryptophan biosynthesis, were significantly enriched (Appendix A).

To assess the relationship between chromatin accessibility and gene expression, ATAC-seq and RNA-seq data were integrated. A total of 1423 motifs identified from ATAC-seq analysis were associated with 285 DEGs (Appendix A). Structural annotation revealed that 910 (63.55%), 188 (13.13%), 109 (7.61%), 107 (7.47%), 99 (6.9%), 14 (0.98%), and five (0.35%) of these motifs were located in distal intergenic, promoter (≤1 kb), promoter (2–3 kb), promoter (1–2 kb), intron, exon, and downstream (≤300 bp) regions, respectively. A TF-gene regulatory network was constructed to predict potential regulatory interactions between motifs and DEGs (Figure 4c). The observed changes in chromatin accessibility linked to differential gene expression likely reflect transcriptional reprogramming during fruit development and ripening. Among the up-regulated motifs in S3 were those associated with AP2/ERF, MYB, zinc finger, NAC, LBD, and WRKY TF families, while motifs corresponding to FAR1 and B3 were down-regulated from S1 to S3.

To verify the accuracy of RNA-sequence data, we selected four significant up-regulated genes, including EVM0003724 (CHS3), EVM0014081 (PAL2), EVM0004311 (MYBPA1), and EVM0019109 (PAL2). The primers were showed in Appendix A. qRT-PCR (quantitative Real-Time Polymerase Chain Reaction) was used to analyze the expression levels of these genes in stage S1 and S3. As shown in Figure 5, the qRT-PCR expression profiles corroborated the transcriptomic data, confirming that all of these genes are up-regulated.

## 3. Discussion

Previous metabolic and transcriptomic analyses have revealed dynamic changes in sugar content and bioactive compound accumulation across developmental stages and among mulberry cultivars [2,4]. While gene expression profiles associated with mulberry fruit development and metabolite biosynthesis have been partially characterized, few studies have explored the role of chromatin remodeling. As a robust method for profiling genome-wide chromatin accessibility, ATAC-seq has been successfully applied in various fruit species [9,11,12]. In this study, an integrated analysis of ATAC-seq and RNA-seq was performed to identify key TFs and their target genes involved in the regulatory processes underlying fruit ripening in the Da10 mulberry cultivar.

Substantial differences in the number of accessible chromatin peaks and FRiP scores were observed between the S1 and S3 stages, with low variation among replicate samples, indicating a high level of data reproducibility. The distribution of THSs differed markedly between stages, with S1 displaying a distinct enrichment near TSS regions (−1 kb to +1 kb) and sharper TSS-centered peaks, suggesting more constrained promoter accessibility during early development. KEGG enrichment analysis of THS-associated genes showed that those specific to S1 were enriched in pathways related to plant growth and development, while S3-specific genes were enriched in pathways related to the biosynthesis of bioactive compounds, such as diterpenoids, flavonoids, and gingerols. These findings are consistent with the role of chromatin accessibility in modulating gene expression during key developmental transitions, such as sex differentiation, growth, and metabolite accumulation [15,18,19].

The greater number of DARs detected in S3 samples suggested a more open chromatin state during the fruit turning stage, potentially reflecting active transcriptional reprogramming. Motif analysis of S1-specific DARs identified five enriched TF families. Among them, TCP factors are implicated in plant growth and mulberry leaf morphology regulation [20]; HMG is reported to regulate plant development and stress tolerance [21,22]; WRKY participates in various biological processes, including development and stress responses [23]; and FAR1 regulates light-responsive processes and adaptation to hypoxic and low-temperature conditions [24]. The enrichment of these motifs in S1 indicates transcriptional programs geared toward early growth and organogenesis in developing fruit. In contrast, S3-specific DARs showed enriched motifs for NAC, MYB, AP2/ERF, and TCP families. NAC TF genes exhibit dynamic transcription changes during mulberry fruit maturation [25], while MYB and AP2/ERF TFs are associated with mulberry fruit development and ripening and are involved in flavonoid biosynthesis and anthocyanin accumulation [1,2,26]. These results correspond with the observed increase in anthocyanin content at S3 (19 days post-anthesis) [2], reflecting active regulation of pigment biosynthesis in late-stage fruit development.

RNA-seq analysis identified significant up-regulation of genes involved in glycolysis/gluconeogenesis, flavonoid biosynthesis, glycine, serine, and threonine metabolism, anthocyanin biosynthesis, and tyrosine, phenylalanine, and tryptophan biosynthesis, consistent with our previous study [2]. qRT-PCR confirmed that the four selected genes were significantly up-regulated at the S3 stage, validating the RNA-seq quantification. Notably, PAL2, MYBPA1, and CHS3 have previously been implicated in flavonoid metabolism and anthocyanin biosynthesis in fruit [27,28]. Integration of ATAC-seq and RNA-seq data revealed that changes in chromatin accessibility were closely correlated with transcriptional regulation during fruit ripening. Although prior studies have noted limited correlation between THS variation and gene expression [29]. Such discrepancies may be due to the influence of additional epigenetic mechanisms—including DNA methylation, histone modification, and non-coding RNA regulation—that shape chromatin structure and transcriptional outcomes [14]. The co-analysis of RNA-seq and ATAC-seq enabled construction of a transcriptional regulatory network, linking chromatin accessibility to differential gene expression [30,31]. In total, 1423 potential TF-binding motifs were associated with 285 DEGs, offering insights into the regulatory architecture governing fruit color development and ripening in mulberry.

## 4. Materials and Methods

### 4.1. Plant Materials

Mulberry varieties were cultivated in the germplasm resource nursery at the Sericulture Research Institute, Sichuan Academy of Agricultural Sciences (Nanchong, China). Fruits were harvested at full maturity; each of three biological replicates comprised six fruits. The complete genome of *Morus atropurpurea* Roxb was downloaded from the China National Center for Bioinformation (CNCB, GWHBOTE00000000) and used as the reference genome. Functional annotation was performed using eggNOG-mapper (v5.0) [32].

### 4.2. Detection of the Content of Anthocyanin in Mulberry Fruit

Total anthocyanins were quantified in 180 fruit samples by the pH-differential spectrophotometric method (Appendix A), with cyanidin-3-O-glucoside (Sigma-Aldrich, Shanghai, China) as the standard [33]. Equipment comprised a Unico spectrophotometer (Shanghai, China), pH meter, sonicator and analytical balance.

### 4.3. ATAC-Seq Library Preparation

The mulberry variety Da10 (*Morus atropurpurea* Roxb, Yueshenda10) was used for ATAC-seq. Four samples were collected at 10 (S1) and 19 (S3) days post-flowering, when the fruits were green (S1) and blushing (S3), respectively (Appendix A). Nuclei suspensions were incubated with a transposition mixture containing transposase, which penetrates the nuclei and selectively fragments DNA in accessible chromatin regions while simultaneously adding sequencing adapters. The transposition reaction was incubated at 37 °C for 30 min. DNA fragments were then purified using a QIAGEN MinElute kit, amplified as described in previous research [34], and sequenced on the Illumina HiSeq^TM^ 4000 platform by Gene Denovo Biotechnology Co., Guangzhou, China.

### 4.4. ATAC Sequencing and Analysis

Adapter sequences, reads containing more than 10% ambiguous nucleotides (N), and those with over 50% low-quality bases (Q-value ≤ 20) were removed from raw data using FASTP (v0.23.2) [35]. Clean reads were aligned to the *M. atropurpurea* Roxb reference genome using BWA (v0.7.17-r1188) [36]. Reads mapping to mitochondrial or chloroplast genomes were filtered out. Duplicate reads were removed using Picard (v3.3.0, https://broadinstitute.github.io/picard/, accessed on 15 October 2025). All reads aligning to the + strand were offset by +4 bps, and all reads aligning to the-strand were offset-5 bps by using deepTools (v3.5.1) [37]. Shifted, properly paired reads were used for peak calling with MACS (v2.1.2) using the parameters “--nomodel--shift-100--extsize 200 B-q 0.05” [38]. The distribution of reads relative to the transcription start site (TSS) position was assessed with deepTools (v3.5.6) [37].

To assess reproducibility between biological replicates, Irreproducible Discovery Rate (IDR) analysis was performed. Peaks with IDR ≤  0.05 were considered reproducible and retained for downstream analysis. Next, the two common peak sets were combined to form a union peak set according to the criteria that individual peaks were merged if overlapping within 300 bp using bedtools (v2.26.0) with the parameter “bedtools multiinter” followed by “bedtools merge -d 300” [39]. The number of fragments within each peak was quantified using htseq-count (v0.9.1) [40]. Differentially accessible peaks between samples were identified using DESeq2 (v1.30.1) with thresholds of |log2 (fold change)| ≥ 1 and false discovery rate (FDR) < 0.05 [41].

Peak-associated genes were identified based on genomic coordinates and gene annotations using ChIPseeker (v1.16.1) [42]. Peak distributions across genomic regions, including promoters, 5′ untranslated regions (UTRs), 3′UTRs, exons, introns, downstream regions, and intergenic regions, were also characterized. Motif discovery was conducted using the MEME Suite (http://meme-suite.org/ accessed on 15 October 2025). MEME-streme was used to identify significantly enriched ungapped motifs, while MEME-Tomtom was used to match discovered motifs against known plant motif databases, including non-redundant plant motifs (https://meme-suite.org/meme/meme-software/Databases/motifs/motif_databases.12.26.tgz accessed on 15 October 2025), JASPAR (https://jaspar.elixir.no/), TAIR (https://www.arabidopsis.org/ accessed on 15 October 2025), and PlantTFDB (https://planttfdb.gao-lab.org/ accessed on 15 October 2025).

### 4.5. Transcription Factor Occupancy Prediction by Footprinting Analysis

TOBIAS (v0.14.0) was employed to predict TF binding sites and assess TF occupancy within differentially accessible regions (DARs) [43]. Footprinting scores were calculated using the ‘FootprintScores’, which estimates signal depletion caused by protein-DNA interactions and evaluates signal patterns flanking the binding sites. Differential TF binding activity was determined using the ‘BINDetect’ module, incorporating TF motifs previously identified by the MEME Suite. TFs with |log2 (fold change in footprinting score)| ≥1 and FDR < 0.05 were considered differentially bound. Footprints from S1 DARs with higher scores in S3, and those from S3 DARs with higher scores in S1, were excluded from further analysis.

### 4.6. Transcriptomic Analysis

RNA-seq data were downloaded from the National Center for Biotechnology Information (NCBI, https://www.ncbi.nlm.nih.gov accessed on 15 October 2025) (accession nos.: SRR11921153, SRR11921152, SRR11921161, SRR11921160, SRR11921159, and SRR11921158). Raw paired-end reads were processed with FASTP (v0.23.2) [30] to remove low-quality sequences and adapters. Clean reads were aligned to the reference genome using STAR (v2.7.10b) [44], and raw read counts were quantified using StringTie (v2.0) [45]. Genes with low expression were removed using the filterByExpr function in edgeR (v4.0.16) [46]. Differentially expressed genes (DEGs) were identified using DESeq2, applying thresholds of |log2 (fold change)| ≥ 1 and adjusted *p*-value ≤ 0.05. By intersecting the differentially expressed genes with the ATAC-seq peak-associated genes, we identified candidate cis-regulatory elements harboring the motif and discussed their putative roles in modulating transcription. To validate the RNA-seq data, total RNA from each sample was reverse-transcribed into cDNA using the HiScript QRT SuperMix for qRT-PCR (+gDNA wiper) (Vazyme, Nanjing, China). EVM0017875 was used as the internal reference gene. Primers for each gene were designed using Primer5. qRT-PCR was performed on a CFX96 instrument (Bio-Rad, Hercules, CA, USA) with FastReal SYBR Green Premix (FP217; Tiangen Biotech, Beijing, China), with reaction conditions of 95 °C for 30 s, 60–63 °C for 30 s, and 72 °C for 1 min over 35 cycles. Each amplification reaction was performed in a 25 µL total volume with 12.5 μL of TB Green Premix Ex Taq II, 1 µL of cDNA and 100 nM (1 μL) of each primer. Gene-expression levels were calculated with the 2^−ΔΔCt^ method [47], and differences between S1 and S3 were assessed with Welch’s *t*-test.

### 4.7. Functional Enrichment and Pathway Analysis

Gene Ontology (GO) and Kyoto Encyclopedia of Genes and Genomes (KEGG) pathway enrichment analyses were conducted using clusterProfiler (v4.10.1) [48]. GO terms were classified into three categories: molecular function (MF), cellular component (CC), and biological process (BP).

## 5. Conclusions

This study provides a comprehensive chromatin accessibility and transcriptomic framework for understanding fruit color development during the turning stage in mulberry. Comparative analysis of ATAC-seq and RNA-seq data between S1 and S3 stages of Da10 fruit revealed distinct transcription factors, gene expression profiles, and regulatory motifs associated with developmental transitions. ATAC-seq identified 1423 motifs corresponding to 23 enriched cis-elements at the S3 stage, including binding sites for the AP2/ERF, MYB, zinc-finger, NAC, LBD and WRKY TF families. By intersecting the ATAC-seq peak-associated genes with differentially expressed genes, 285 DEG that may be regulated during fruit ripening by transcription factors. Intersection of ATAC-seq peak-associated genes with the differentially expressed set identified 285 DEGs that are candidate targets of transcription factors regulating fruit color change. One CHS3, one MYBPA1, and two PAL2 genes were validated by qRT-PCR. These findings establish a foundation for future investigations into the epigenomic control of fruit maturation. Nonetheless, functional validation through targeted genetic and biochemical assays remains essential to confirm the role of chromatin accessibility in regulating pigment accumulation during mulberry fruit development.

## Figures and Tables

**Figure 1 ijms-27-00456-f001:**
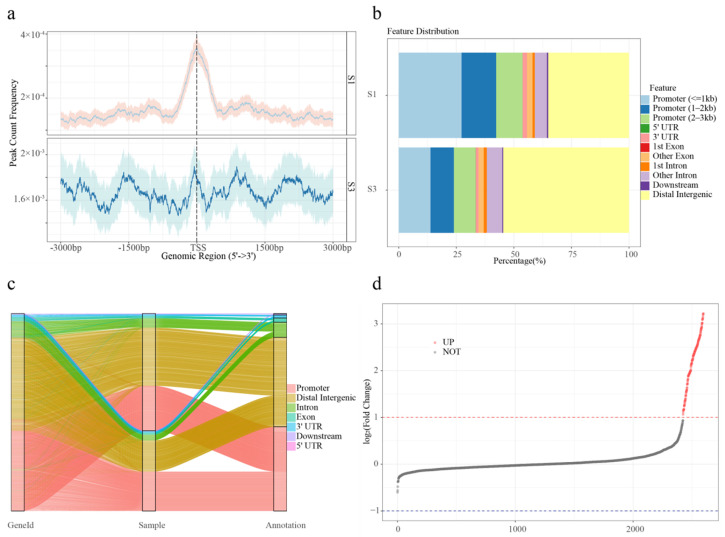
Chromatin accessibility landscape of S1 and S3 samples. (**a**) Average plots showing the signals at the TSSs in the ATAC-seq data sets. (**b**) Feature distributions of THSs of S1 and S3. (**c**) Sankey diagram showing the changes in the proportion of gene features of S1 and S3. (**d**) Differentially accessible regions (DARs) are showed in point plot. Red dots indicate significant up-regulation; black dots indicate no significant difference.

**Figure 2 ijms-27-00456-f002:**
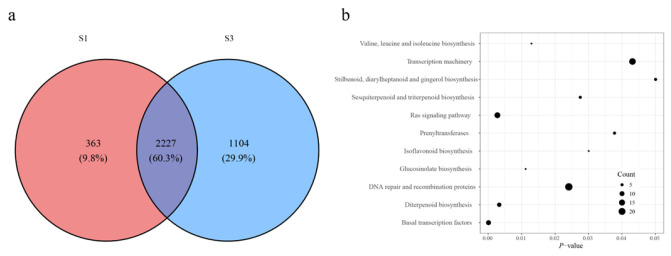
Functional analysis of the genes adjacent to THSs and DARs. (**a**) Venn diagram showing the shared genes between two ATAC-seq data sets. (**b**) Pathway enrichment analysis of the S3-specific gene set.

**Figure 3 ijms-27-00456-f003:**
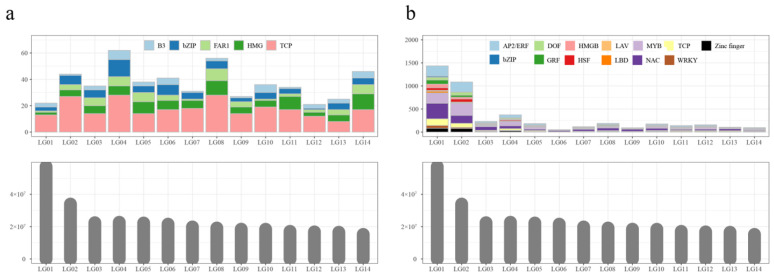
The chromosomal distribution of the motifs of S1 (**a**) and S3 (**b**) samples.

**Figure 4 ijms-27-00456-f004:**
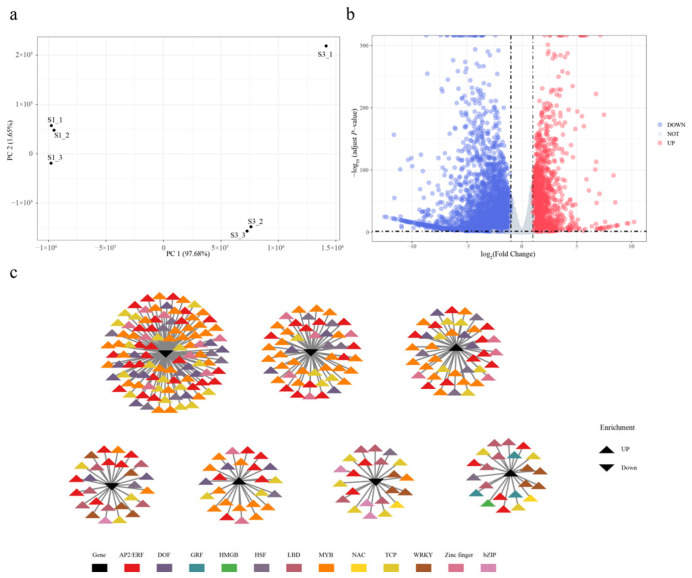
Integrated analysis of RNA-seq and ATAC-seq data. (**a**) PCA analysis of six individuals. (**b**) Up- and down-regulation of differential gene signal. Genes marked in red are significantly up-regulated genes and those marked in blue are significantly down-regulated genes. (**c**) Co-expression network of differentially expressed genes connected with differential ATAC signaling motif enrichment transcription factors. The triangles in black represent genes and the colored triangles represent ATAC signaling motif enrichment transcription factors that are associated with these genes.

**Figure 5 ijms-27-00456-f005:**
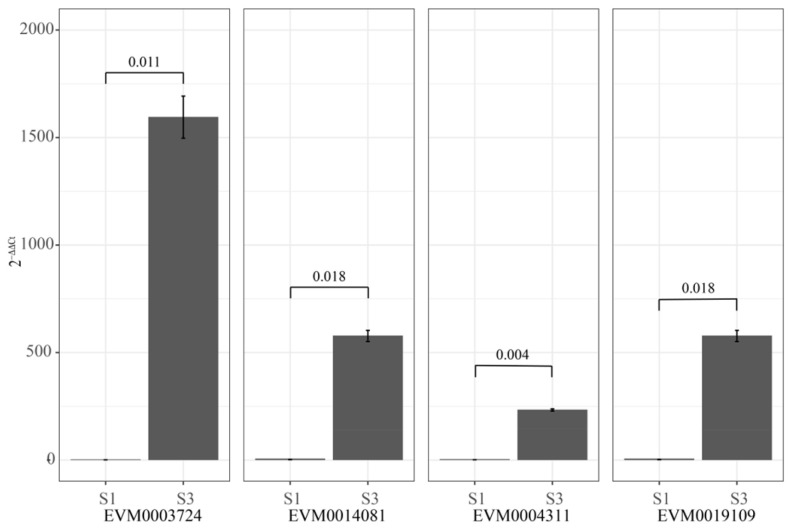
Comparison of expression value measured by RNA-seq and qRT-PCR. The 2^−ΔΔCt^ values for EVM0003724 (CHS3), EVM0014081 (PAL2), EVM0004311 (MYBPA1), and EVM0019109 (PAL2) were used to quantify gene expression levels. The significant difference was evaluated by Welch Satterthwaite *t*-test.

## Data Availability

The raw reads of ATAC-seq of four Da10 mulberry were deposited in the Genome Sequence Archive (Genomics, Proteomics and Bioinformatics 2021) in National Genomics Data Center (NGDC, https://ngdc.cncb.ac.cn/ accessed on 15 October 2025), China National Center for Bioinformation/Beijing Institute of Genomics, Chinese Academy of Sciences (PRJCA044543). The complete genome of *M. atropurpurea* Roxb was downloaded from NGDC with accession number GWHBOTE00000000. RNA-seq data were downloaded from the National Center for Biotechnology Information (NCBI, https://www.ncbi.nlm.nih.gov accessed on 15 October 2025) (accession nos.: SRR11921153, SRR11921152, SRR11921161, SRR11921160, SRR11921159, and SRR11921158).

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
