# Peer review of "ATAC-Seq and RNA-Seq Integration Reveals Chromatin Accessibility and Transcriptional Dynamics During Fruit Color Development in Mulberry"

_ijms, 2026, doi:10.3390/ijms27010456_

Round 1
Reviewer 1 Report
Comments and Suggestions for Authors
The authors integrated the ATAC-seq and RNA-seq data to reveal information in mulberry. Here are my suggestions:
Abstract.
Include information about the qPCR results.
Introduction
L65. Please add a dot after the reference “13”.
L67-L69. Please edit the paragraph to improve the meaning. Usually, the word “when” is not used after a dot.
L71. I suggest that the authors should explain briefly the transitional stages (S1 and S3). Why are these important? What are these stages? Is there any document that classifies these stages? Please include all this information to improve the meaning.
Results
L92. M. atropurpurea genome (L92, L157) is the same of Morus genome (L131)? Please review this and use only one term or explain it.
L231. Please include the full abbreviation of RT-PCR (real time…)
I suggest improve the figure 4 and figure 5. It seems low quality figures.
L236. In Figure 5 (please add a space after figure). The authors should perform a statistical analysis, or at least a correlation analysis to prove that RNA-seq and RT-PCR results are similar.
Discussion
L280-282. The authors stated that they integrated the ATAC and RNA-seq data, observing changes in chromatin accessibility (association). However, the next line is focus on the correlation analysis. In this regard, association and correlation are two different things. Please edit the paragraph to improve the meaning.
L288. Please review the number 1 423… are 1,423 or only 423? Please review it.
In general, the discussion is too short.
What about the RT-PCR vs RNA-seq results? Are this correlated? Are this associated?
Are the differences? Please include a discussion about the RT-PCR and if it possible, a statistical analysis.
Materials and methods
L293. Please include the name of all the mulberry varieties used. Moreover, the authors should specify why they selected the da10 variety for the subsequent analysis..
What do you mean with “defined developmental stages”??. Please include more detailed information due to the material section is at the end of the manuscript.
L301-305. Please improve the meaning of the paragraph, is too confusing.
L367. Please use qPCR or RT-PCR, only one across the manuscript.
L368. Please include the sequence of the primers used.
L368-371. Authors need to include the qPCR reaction and specify the quantities used for the PCR and for the qPCR. (i.e. 1 uL of Fw primer, 1 uL Rv primer, ng of cDNA, SYBR GREEN.. etc)
On the other hand, authors stated that performed a qPCR or a RT-PCR, and only mention a T100 Thermal cycler. The thermocycler mentioned cannot perform a qPCR. Authors need to explain and difference between the conventional PCR and the qPCR.
Furthermore, authors should report the reference gene and sample calibrator as well as the formula to calculate the values reported. Please include all this information.
Please also include a statistical design (for the qPCR analysis)
On the other hand, there is a lack of information on how the authors integrate both dat (ATC and RNA-seq). The authors should explain the bioinformatic analysis performed and include it before the conclusion section.
Conclusions.
Authors need to improve the conclusion section. I suggest reporting the most important findings of the study. What are the most important genes? What are they associated with? Co-expression network? Genes associated? What about the qPCR results?
Author Response
The authors integrated the ATAC-seq and RNA-seq data to reveal information in mulberry. Here are my suggestions:
Abstract.
Include information about the qPCR results.
Response: Thanks for your suggestion. Revised as suggested.
Introduction
L65. Please add a dot after the reference “13”.
Response: Thanks for your suggestion. We have revised.
L67-L69. Please edit the paragraph to improve the meaning. Usually, the word “when” is not used after a dot.
Response: Thanks for your comments. This paragraph was revised.
L71. I suggest that the authors should explain briefly the transitional stages (S1 and S3). Why are these important? What are these stages? Is there any document that classifies these stages? Please include all this information to improve the meaning.
Response: Thanks for your comments. As illustrated in Supplementary Figure S9 and detailed in the Methods, S1 and S3 correspond to 10 and 19 days after flowering of mulberry, respectively, with fruit colored green and blushing.
Results
L92. M. atropurpurea genome (L92, L157) is the same of Morus genome (L131)? Please review this and use only one term or explain it.
Response: Thanks for your comments. These genomes are identical; we have revised them in the manuscript.
L231. Please include the full abbreviation of RT-PCR (real time…)
Response: Revised as suggested.
I suggest improve the figure 4 and figure 5. It seems low quality figures.
Response: Thanks for your comments. We will supply the figure as a 500-dpi image and as vector graphics (PDF).
L236. In Figure 5 (please add a space after figure). The authors should perform a statistical analysis, or at least a correlation analysis to prove that RNA-seq and RT-PCR results are similar.
Response: Thanks for your suggestion. We have revised.
Figure 5. Comparison of expression value measured by RNA-seq and qRT-PCR. The 2-ΔΔCt values for EVM0003724 (CHS3), EVM0014081 (PAL2), EVM0004311 (MYBPA1), and EVM0019109 (PAL2) were used to quantify gene expression levels. The significant difference was evaluated by Welch Satterthwaite t-test.
Discussion
L280-282. The authors stated that they integrated the ATAC and RNA-seq data, observing changes in chromatin accessibility (association). However, the next line is focus on the correlation analysis. In this regard, association and correlation are two different things. Please edit the paragraph to improve the meaning.
Response: Thanks for your suggestion. These typos were revised.
L288. Please review the number 1 423… are 1,423 or only 423? Please review it.
Response: Thank you for your comment. The correct number is 1,423, and we have revised it accordingly.
In general, the discussion is too short.
What about the RT-PCR vs RNA-seq results? Are this correlated? Are this associated?
Are the differences? Please include a discussion about the RT-PCR and if it possible, a statistical analysis.
Response: Thank you for the suggestion. The qRT-PCR data have been integrated with the RNA-seq results in the Discussion, and Welch’s t-test was used to compare gene-expression levels between S1 and S3 stage.
Materials and methods
L293. Please include the name of all the mulberry varieties used. Moreover, the authors should specify why they selected the da10 variety for the subsequent analysis..
Response: Most of the artificially bred varieties lack formal Latin names; we have therefore provided their common names instead (Supplementary Table S1). We selected the Da10 cultivar (Morus atropurpurea Roxb.) for downstream analysis because it accumulates the highest anthocyanin levels in fresh fruit (Section: Results) and is the most valued mulberry fruit for functional-component content (Song, W., (2009)).
What do you mean with “defined developmental stages”??. Please include more detailed information due to the material section is at the end of the manuscript.
Response: Thanks for your suggestion. This was revised.
L301-305. Please improve the meaning of the paragraph, is too confusing.
Response: Thanks for your suggestion. This paragraph has been revised accordingly.
L367. Please use qPCR or RT-PCR, only one across the manuscript.
Response: Revised as suggested.
L368. Please include the sequence of the primers used.
Response: Thank you for the suggestion; the information has been added to Table R1 and Supplementary Table S9.
Table R1. Primer sequence and product length of the target genes.
|
Gene ID |
Forward |
Reverse |
Product (bp) |
Tm (℃) |
|
EVM0003724 |
GGCTAAGTGTGGCTTGAA |
TCCTTCTCCTGTGGTCTT |
151 |
61 |
|
EVM0014081 |
AGCACCTCAATCTTCCAA |
ACCTTCTCACCTGTTAGC |
200 |
60 |
|
EVM0019109 |
AGCACCTCAATCTTCCAA |
ACCTTCTCACCTGTTAGC |
200 |
60 |
|
EVM0004311 |
CATCTCCCGAAACCCATTAG |
GCCATCATCACCGACTTC |
174 |
62 |
|
EVM0017875 |
TAGCAGCACAAGCCTCAG |
AAGAAGCAGCAGCCAGAG |
179 |
63.5 |
L368-371. Authors need to include the qPCR reaction and specify the quantities used for the PCR and for the qPCR. (i.e. 1 uL of Fw primer, 1 uL Rv primer, ng of cDNA, SYBR GREEN.. etc)
On the other hand, authors stated that performed a qPCR or a RT-PCR, and only mention a T100 Thermal cycler. The thermocycler mentioned cannot perform a qPCR. Authors need to explain and difference between the conventional PCR and the qPCR.
Response: Thank you for the suggestion; this information has been added to the manuscript (Section: Method).
Furthermore, authors should report the reference gene and sample calibrator as well as the formula to calculate the values reported. Please include all this information.
Please also include a statistical design (for the qPCR analysis)
Response: Thank you for the suggestion. Revised as suggested.
On the other hand, there is a lack of information on how the authors integrate both dat (ATC and RNA-seq). The authors should explain the bioinformatic analysis performed and include it before the conclusion section.
Response: Thank you for the comment. We have revised it as “By intersecting the differentially expressed genes with the ATAC-seq peak-associated genes, we identified candidate cis-regulatory elements harboring the motif and discuss their putative roles in modulating transcription.”
Conclusions.
Authors need to improve the conclusion section. I suggest reporting the most important findings of the study. What are the most important genes? What are they associated with? Co-expression network? Genes associated? What about the qPCR results?
Response: Thank you for the comment; the Conclusion section has been expanded as suggested.
Reference
Song, W., Wang, H. J., Bucheli, P., Zhang, P. F., Wei, D. Z., Lu, Y. H. Phytochemical profiles of different mulberry (Morus spp.) species from China. Journal of agricultural and food chemistry 2009, 57, (19), 9133-9140.

Reviewer 2 Report
Comments and Suggestions for Authors
This manuscript describes the use of an integrated approach combining ATAC-seq and RNA-seq to reconstruct the regulatory network governing mulberry fruit coloration during growth (S1 stage) and ripening (S3 stage). Based on these data, it is possible to identify candidate genes playing important roles in the accumulation of key metabolites, which can be used in mulberry breeding programs.
In general, the manuscript can be accepted for publication in IJMS after some corrections.
In Introduction:
L39: Fruit development involves – Fleshy fruit development involves
L42: “…such as anthocyanins [1, 4-6].” – …or carotenoids in case of tomato [5]
L43-45: In the study [8], the main attention is paid to the analysis of the influence on the anthocyanin synthesis of the abnormal expression of the bHLH3 gene, which encodes a transcription factor involved in the regulation of the anthocyanin synthesis genes transcription, and not a structural gene.
L45-48: In addition to ethylene-responsive factors, there are many other regulatory genes that influence the ripening of fleshy fruits. For example, a review of blueberry ripening provides lists of genes (both structural and regulatory) involved in the regulatory network of different stages of berry ripening [Zapien-Macias et al. Blueberry ripening mechanism: a systematic review of physiological and molecular evidence. Hortic. Res. 2025;12(8):uhaf126. doi: 10.1093/hr/uhaf126].
L56-61: It would be good to write here that the TFs involved in the regulation of anthocyanin synthesis are also known - this is the MBW complex, which has been described in detail and combines three TFs from different families (MYB, bHLH, WD40) [for example, Zhao et al. Identification and Characterization of MYB-bHLH-WD40 Regulatory Complex Members Controlling Anthocyanidin Biosynthesis in Blueberry Fruits Development. Genes (Basel). 2019;10(7):496. doi: 10.3390/genes10070496.].
Results:
L81-82: ‘Da10’ is Yueshenda10 (Suppl. Table 1), right? Then add a clarification in the text that you shortened the name for simplicity. At what stage of the fruit was the anthocyanin content measured?
L86: ‘data were generated from four samples’ – ‘data were generated from four fruit samples collected at 10 (S1) and 19 (S3) days after flowering’. (Some details are needed as materials and methods are located at the end of the manuscript.)
L92, 157, 296: M. atropurpurea - M. atropurpurea
L115: Figure. 1C should be Figure. 1B? On Fig. C it is not clear where S1 is and where S3 is. The drawing is blind and blurry, the captions are not visible even when enlarged, and the Figure caption is insufficient in explanation.
L136-137: ‘The 173 DARs up-regulated in S3 were located near 92 genes (Supplementary Figure. S6).’ Additional Fig. 6 shows Venn diagrams for S1/S3-specific genes, divided by THS localization. It is unclear where to look to see the 173 upregulated DARs near 92 genes. Please clarify.
L165: ‘These motifs varied from 6 to 50 amino acids in length.’ - The authors mean nucleotide sequences, not amino acids, right?
L165-168: The link to additional table 5 should probably be after the first (no second) of the two sentences, right? – ‘Alignment with known motif databases identified 1 014 NAC, 793 MYB, 713 AP2/ERF, and 413 TCP motifs (Supplementary Table S5)’. And ‘Chromosomal distribution analysis revealed substantial variation across the 14 chromosomes (Figure 3A).’
L186-187: ‘RNA-seq was conducted using three biological replicates for each stage (Supplementary Figure. S9).’ - It is necessary to write this text according to the chapter on materials and methods that “RNA-seq data for S1 and S3 stages of fruit maturation in three biological replicates were downloaded from the NCBI database and processed bioinformatically”.
L187: Perhaps it is also necessary to refer to supplementary table S2.
L190: M. notabilis - M. notabilis
Figure 4C: Please explain in the caption to the figure which network of the seven corresponds to what (according to lines 219-221?).
L230-231: Supplementary Table S7 - Supplementary Table S8
L231: Here (and throughout the text), qRT-PCR (= quantitative real-time PCR) should be used.
L228-230: It would be good to clarify here that all four genes belong to the flavonoid pathway (and, in particular, the anthocyanin pathway).
The important question about the Results is the following. In total, the authors were able to map 1,423 motifs identified by ATAC-seq to 285 DEGs from RNA-seq. Please provide a list of these genes (including their annotation, accession numbers and ‘S3 vs S1’ RPKM changing (up- or down-regulation)) as a supplementary table so that readers can also draw conclusions about the participants in the regulatory network that determines mulberry fruit ripening. If I understand correctly, this will be a list of target genes, transcription of which is regulated during fruit ripening by transcription factors of the TCP, HMG, bZIP, FAR1, B3, etc. families. Again, how many and which of these 285 genes are related to flavonoid/anthocyanin biosynthesis, and how many/which ones are related to carbohydrate metabolism. This will give cognitive biological meaning to the bioinformatically processed results. In addition, to understand the regulation of the target process, it would be interesting to supplement the ‘target genes’ results with RNA-seq data for specific TFs of the indicated families, the genes of which significantly changed expression in S3 versus S1. Also, were among the differentially expressed TFs those known to regulate flavonoid biosynthesis genes according to [Zhao et al. doi: 10.3390/genes10070496.], for example?
Material and Methods:
Section 4.2 - Describe or provide a link to a description of the method for determining the content of anthocyanins in plant tissue.
Author Response
This manuscript describes the use of an integrated approach combining ATAC-seq and RNA-seq to reconstruct the regulatory network governing mulberry fruit coloration during growth (S1 stage) and ripening (S3 stage). Based on these data, it is possible to identify candidate genes playing important roles in the accumulation of key metabolites, which can be used in mulberry breeding programs.
In general, the manuscript can be accepted for publication in IJMS after some corrections.
In Introduction:
L39: Fruit development involves – Fleshy fruit development involves
Response: Thank you for the suggestion. Revised as suggested.
L42: “…such as anthocyanins [1, 4-6].” – …or carotenoids in case of tomato [5]
Response: Thank you for the suggestion. Revised as suggested.
L43-45: In the study [8], the main attention is paid to the analysis of the influence on the anthocyanin synthesis of the abnormal expression of the bHLH3 gene, which encodes a transcription factor involved in the regulation of the anthocyanin synthesis genes transcription, and not a structural gene.
Response: Thank you for the suggestion. We have revised it as “Most previous studies have focused on a narrow set of structural genes or transcription factor involved in metabolic biosynthesis, offering an incomplete view of regulatory dynamics [7, 8].”.
L45-48: In addition to ethylene-responsive factors, there are many other regulatory genes that influence the ripening of fleshy fruits. For example, a review of blueberry ripening provides lists of genes (both structural and regulatory) involved in the regulatory network of different stages of berry ripening [Zapien-Macias et al. Blueberry ripening mechanism: a systematic review of physiological and molecular evidence. Hortic. Res. 2025;12(8):uhaf126. doi: 10.1093/hr/uhaf126].
Response: Thank you for the comment. We fully agree that numerous regulatory genes control fleshy-fruit ripening. Here, we specifically focused on the regulatory genes that drive mulberry fruit ripening.
L56-61: It would be good to write here that the TFs involved in the regulation of anthocyanin synthesis are also known - this is the MBW complex, which has been described in detail and combines three TFs from different families (MYB, bHLH, WD40) [for example, Zhao et al. Identification and Characterization of MYB-bHLH-WD40 Regulatory Complex Members Controlling Anthocyanidin Biosynthesis in Blueberry Fruits Development. Genes (Basel). 2019;10(7):496. doi: 10.3390/genes10070496.].
Response: Thank you for the suggestion. Revised as suggested and this reference have been added.
Results:
L81-82: ‘Da10’ is Yueshenda10 (Suppl. Table 1), right? Then add a clarification in the text that you shortened the name for simplicity. At what stage of the fruit was the anthocyanin content measured?
Response: Thank you for the suggestion. Da10 corresponds to “Yueshenda10” in Supplementary Table 1, and anthocyanin content was quantified at full maturity; this information has been added to the manuscript (Section: Method).
L86: ‘data were generated from four samples’ – ‘data were generated from four fruit samples collected at 10 (S1) and 19 (S3) days after flowering’. (Some details are needed as materials and methods are located at the end of the manuscript.)
Response: Thank you for the suggestion. Revised as suggested.
L92, 157, 296: M. atropurpurea - M. atropurpurea
Response: Thank you for the suggestion. Revised as suggested.
L115: Figure. 1C should be Figure. 1B? On Fig. C it is not clear where S1 is and where S3 is. The drawing is blind and blurry, the captions are not visible even when enlarged, and the Figure caption is insufficient in explanation.
Response: Thank you for your comment. In Figure 1B, the misaligned baselines of the stacked bars could mislead readers, whereas Figure 1C (the Sankey diagram) unambiguously shows how the total pool of re-presented genes is partitioned into distinct categories. And we have resubmitted higher-resolution images.
L136-137: ‘The 173 DARs up-regulated in S3 were located near 92 genes (Supplementary Figure. S6).’ Additional Fig. 6 shows Venn diagrams for S1/S3-specific genes, divided by THS localization. It is unclear where to look to see the 173 upregulated DARs near 92 genes. Please clarify.
Response: Thank you for your comment. We apologize for the imprecise wording. Our intended message is: Figure 2a depicts the global association between THSs and genes; Then these data are integrated with the DARs described in Section 2.2, and 92 genes are identified. These relationships are detailed in Figure R1. And this sentence was revised as “Among them, 173 S3-up-regulated DARs mapped adjacent to 92 genes.”
Figure R1. Venn diagram displaying genes shared among S1, S3 and DAR-associated sets.
L165: ‘These motifs varied from 6 to 50 amino acids in length.’ - The authors mean nucleotide sequences, not amino acids, right?
Response: Thanks for your suggestion. These typos were revised.
L165-168: The link to additional table 5 should probably be after the first (no second) of the two sentences, right? – ‘Alignment with known motif databases identified 1 014 NAC, 793 MYB, 713 AP2/ERF, and 413 TCP motifs (Supplementary Table S5)’. And ‘Chromosomal distribution analysis revealed substantial variation across the 14 chromosomes (Figure 3A).’
Response: Yes, you are absolutely right—thank you for catching that. The link to Supplementary Table S5 should indeed follow the first sentences. The sentence “Chromosomal distribution analysis revealed substantial variation across the 14 chromosomes” should cite Figure 3B, mirroring the reference to Figure 3A for the S1 motifs.
L186-187: ‘RNA-seq was conducted using three biological replicates for each stage (Supplementary Figure. S9).’ - It is necessary to write this text according to the chapter on materials and methods that “RNA-seq data for S1 and S3 stages of fruit maturation in three biological replicates were downloaded from the NCBI database and processed bioinformatically”.
Response: Thank you for the suggestion; we have revised the manuscript accordingly.
L187: Perhaps it is also necessary to refer to supplementary table S2.
Response: Revised as suggested.
L190: M. notabilis - M. notabilis
Response: Thanks for your suggestion. This typo has been corrected.
Figure 4C: Please explain in the caption to the figure which network of the seven corresponds to what (according to lines 219-221?).
Response: Thank you for your comment. Figure 4C displays the seven networks whose associated motifs exhibited the highest connectivity.
L230-231: Supplementary Table S7 - Supplementary Table S8
Response: Thanks for your suggestion. This typo has been corrected.
L231: Here (and throughout the text), qRT-PCR (= quantitative real-time PCR) should be used.
Response: Thank you for the suggestion. Revised as suggested.
L228-230: It would be good to clarify here that all four genes belong to the flavonoid pathway (and, in particular, the anthocyanin pathway).
Response: Thank you for the suggestion. We have incorporated this information into the Discussion section.
The important question about the Results is the following. In total, the authors were able to map 1,423 motifs identified by ATAC-seq to 285 DEGs from RNA-seq. Please provide a list of these genes (including their annotation, accession numbers and ‘S3 vs S1’ RPKM changing (up- or down-regulation)) as a supplementary table so that readers can also draw conclusions about the participants in the regulatory network that determines mulberry fruit ripening. If I understand correctly, this will be a list of target genes, transcription of which is regulated during fruit ripening by transcription factors of the TCP, HMG, bZIP, FAR1, B3, etc. families. Again, how many and which of these 285 genes are related to flavonoid/anthocyanin biosynthesis, and how many/which ones are related to carbohydrate metabolism. This will give cognitive biological meaning to the bioinformatically processed results. In addition, to understand the regulation of the target process, it would be interesting to supplement the ‘target genes’ results with RNA-seq data for specific TFs of the indicated families, the genes of which significantly changed expression in S3 versus S1. Also, were among the differentially expressed TFs those known to regulate flavonoid biosynthesis genes according to [Zhao et al. doi: 10.3390/genes10070496.], for example?
Response: Thank you for the suggestion. The gene list is now provided as Supplementary Table S9, and the previous Supplementary Table S8 has been renumbered accordingly.
Material and Methods:
Section 4.2 - Describe or provide a link to a description of the method for determining the content of anthocyanins in plant tissue.
Response: Thank you for the suggestion. Revised as suggested.
Reference
Lee J., Durst R W., Wrolstad R E. Determination of total monomeric anthocyanin pigment content of fruit juices, beverages, natural colorants, and wines by the pH differential method: collaborative study. Journal of AOAC international 2005, 88, (5), 1269-1278.

Round 2
Reviewer 1 Report
Comments and Suggestions for Authors
The authors performed the changes requested. Article was improved and can be accepted.